# The Association between Metabolically Healthy Obesity, Cardiovascular Disease, and All-Cause Mortality Risk in Asia: A Systematic Review and Meta-Analysis

**DOI:** 10.3390/ijerph17041320

**Published:** 2020-02-19

**Authors:** Ming-Yuan Huang, Mu-Yi Wang, Yu-Sheng Lin, Chien-Ju Lin, Kai Lo, I-Jen Chang, Ting-Yao Cheng, Szu-Ying Tsai, Hsin-Hao Chen, Chien-Yu Lin, Shu Jung Liu, Kuo-Liong Chien, Tzu-Lin Yeh

**Affiliations:** 1Department of Family Medicine, Taipei MacKay Memorial Hospital, No. 92, Section 2, Zhongshan North Road, Taipei City 10449, Taiwan; martin.h912@gmail.com (M.-Y.H.); b98401116@g.ntu.edu.tw (S.-Y.T.); 2Department of Medicine, MacKay Medical College, No. 46, Sec. 3, Zhongzheng Rd., Sanzhi Dist., New Taipei City 25245, Taiwan; 2033@mmh.org.tw (H.-H.C.); mmhped.lin@gmail.com (C.-Y.L.); 3Hospice and Palliative Care Center, MacKay Memorial Hospital, No. 45, Minsheng Rd., Tamsui District, New Taipei City 25160, Taiwan; 4Post Graduate Year, Taipei MacKay Memorial Hospital, No. 92, Section 2, Zhongshan North Road, Taipei City 10449, Taiwan; vlentry@gmail.com (M.-Y.W.); u102001005@cmu.edu.tw (Y.-S.L.); u102001791@cmu.edu.tw (K.L.); evanchang0909@gmail.com (I.-J.C.); bill8471bill@gm.ym.edu.tw (T.-Y.C.); 5Department of Family Medicine, Hsinchu MacKay Memorial Hospital, No. 690, Section 2, Guangfu Road, East District, Hsinchu City 30071, Taiwan; tttyii213@gmail.com; 6MacKay Junior College of Medicine, Nursing, and Management, No. 92, Shengjing Rd., Beitou Dist., Taipei City 11260, Taiwan; 7Department of Pediatrics, Hsinchu MacKay Memorial Hospital, Hsinchu 30071, Taiwan; 8Department of Medical Library, MacKay Memorial Hospital, No. 45, Minsheng Rd., Tamsui District, New Taipei City 25160, Taiwan; mmhlibrarian@gmail.com; 9Department of Internal Medicine, National Taiwan University Hospital, No. 7, Zhongshan S. Rd., Zhongzheng Dist., Taipei City 10002, Taiwan; klchien@ntu.edu.tw; 10Institute of Epidemiology and Preventive Medicine, National Taiwan University, Room 517, No.17, Xu-Zhou Rd., Taipei City 10055, Taiwan

**Keywords:** metabolically healthy obesity, cardiovascular disease, all-cause mortality, Asia, meta-analysis

## Abstract

We investigated the association among metabolically healthy obesity (MHO), cardiovascular disease (CVD)risk, and all-cause mortality in the Asian population. We searched databases from inception to 16 November, 2019 and pooled data using a random-effects model. Subgroup analysis was conducted according to the following comparison groups: MHNW (without overweight or underweight participants) and MHNO (non-obese, including overweight and underweight participants). Nineteen studies were included. The mean Newcastle–Ottawa Scale score was 7.8. Participants with MHO had a significantly higher CVD risk (odds ratio (OR) = 1.36, 95% confidence interval (CI) = 1.13–1.63) and significantly lower risk of all-cause mortality (OR = 0.88, 95% CI = 0.78–1.00) than the comparison group. Subgroup analyses revealed participants with MHO had a significantly higher CVD risk than MHNW participants (OR = 1.61; 95% CI = 1.24–2.08; I^2^ = 73%), but there was no significant difference compared with MHNO participants (OR, 1.04; 95% CI, 0.80–1.36; I^2^ = 68%). Participants with MHO had a significantly lower risk of all-cause mortality (OR = 0.83; 95% CI = 0.78–0.88; I^2^ = 9%) than MHNO participants, but a borderline significantly higher risk of all-cause mortality than MHNW participants (OR = 1.30; 95% CI = 0.99–1.72; I^2^ = 0%). The CVD risk and all-cause mortality of the MHO group changed depending on the control group. Thus, future studies should select control groups carefully.

## 1. Introduction

Cardiovascular disease (CVD) is an important global health issue. In 2016, an estimated 17.9 million people died from CVD, representing 31% of all global deaths [1]. In Asia, the population of obese individuals with metabolic syndrome has increased dramatically [2]. CVD is a significant health issue in Asian society and is the leading cause of death among Asians [3].

Although all obesity phenotypes do not share the same characteristics, obesity has been recognized as a major health problem worldwide [4]. Metabolically healthy obesity (MHO) has drawn increasing attention in recent years and has been reported to have a better cardiovascular and metabolic outcome than metabolically unhealthy obesity [5]. However, the definition of MHO is not uniform; it has been primarily described as obesity without any metabolic abnormalities or obesity without metabolic syndrome with less than one or two of the five metabolic traits (high blood pressure, plasma triglyceride concentration, or fasting blood glucose; low high-density lipoprotein cholesterol concentration; and large waist circumference) [6]. 

Several cohort studies have discussed the relationship between MHO and CVD risk and all-cause mortality. While recent systematic reviews have confirmed the relationship between MHO and increased CVD risk and all-cause mortality compared with “metabolically healthy normal weight (MHNW)” [7,8], most of these studies included only Caucasian populations. Furthermore, in several Korean studies that compared MHO individuals with “metabolically healthy non-obese (MHNO)”, individuals are classified as obese or non-obese, and data on overweight and underweight individuals were excluded from these meta-analyses [5,9]. Multivariate meta-analysis in previous research disclosed that the increasing age and proportion of women in MHO group led to the decline of CVD risk, which indicated that age and gender may be important effect modifiers [7]. Therefore, this research tried to survey if a similar effect can be found among the Asian group. As ethnicity is an independent and important confounding factor related to both anthropometrics and CVD risk, this is a fatal flaw of the aforementioned reviews [6]. Body mass index (BMI) has been reported to be the most common index of obesity. However, body composition and fat distribution are different between Asians and Caucasians. Therefore, obesity, as defined by BMI, might be underestimated in the Asian population [10]. Thus, our aim was to perform a comprehensive systematic review and meta-analysis to investigate the relationship between MHO and CVD and all-cause mortality in the Asian population and examine the difference with respect to the control groups of metabolically healthy normal weight (MHNW) and MHNO individuals.

## 2. Methods 

### 2.1. Search Strategy and Study Eligibility

We designed and conducted this review with reference to the Preferred Reporting Items for Systematic Reviews and Meta-Analyses (PRISMA; Appendix A) [11]. We searched electronic databases (PubMed/Medline, Cochrane Database of Systematic Review, the Cochrane Central Register of Controlled Trials, Embase and Cumulative Index to Nursing, and Allied Health Literature) and scanned the reference lists of the related articles manually for articles published before 16 November, 2019. Our primary and secondary outcomes were CVD risk and all-cause mortality, respectively. We selected articles that compared CVD risk and all-cause mortality of MHO with MHNO and MHNW in the Asian population. We searched all article types except letters and editorials for ascertaining the quality of the methodology and did not limit the language during the search. Studies were included if they met the following criteria: (1) The study cohort only recruited Asian participants who were 18 years or older; (2) the research interest was MHO; (3) comparisons were made with MHNO and MHNW participants; (4) the study outcomes met the composite definitions of CVD (including coronary heart disease, stroke, and heart failure) and all-cause mortality; and (5) the study adopted a cohort study design. Full search strategies are shown in Appendix A. We excluded duplicated articles, articles not relevant to MHO, studies that did not investigate the Asian population, studies that had outcomes other than CVD, cross-sectional studies, and review articles. Two reviewers independently screened the titles, abstracts, as well as full-text papers to determine eligibility, and disagreements were resolved by discussion with a third reviewer.

### 2.2. Data Extraction and Quality Assessment

A standardized data collection form was implemented to extract data. We extracted the following information from each study: First author’s name, year of publication, country where the cohort study was conducted, numbers of participants, proportion of women, percentage of smokers, prevalence of MHO, participant age at baseline, follow-up duration, definition of obesity, BMI categories (kg/m^2^), definitions of metabolically healthy/unhealthy status, adjusted variables, definition of CVD, and main results from the tables and text of the full paper. Adjusted odds ratio (OR) for the risk of CVD and all-cause mortality were extracted from each study.

We assessed the methodological quality of all studies using the Newcastle–Ottawa Assessment Scale (NOS) for cohort studies [12]. The NOS assigns a maximum of nine stars to studies with the least risk of bias, four stars for selection, two stars for comparability, and three stars for ascertainment. Two independent reviewers completed the assessment, and any disagreements were resolved by a third reviewer.

All statistical analyses and plotting were performed using the meta package and metafor package in the RStudio software (version 1.2.1335; RStudio, Inc., Boston, MA, USA). For the meta-analysis of cohort studies, OR was transformed to natural logarithms and the 95% confidence intervals (CIs) were used to calculate the standard errors. If multiple effect sizes were reported in one study, we calculated the average logarithms of the OR (LOR) and the average standard errors. We assumed that the true effect could vary between studies. Thus, we employed a random-effects model. We used the I^2^ statistic and Cochran’s Q-test to measure heterogeneity. A significant Q test (*p*-value < 0.05) or I^2^ > 50% indicated the presence of heterogeneity [13]. To address heterogeneity, we carried out subgroup analyses according to the following comparison groups: MHNW (without overweight or underweight participants) and MHNO (non-obese, including overweight and underweight participants). If more than 10 articles were included in the meta-analysis, we performed meta-regression to investigate potential effect modifiers and origins of heterogeneity. Sensitivity analyses were conducted by omitting each study to test the stability of the results and removing studies with a mean follow-up duration of fewer than five years to prevent the possibility of an inverse causal relationship. Finally, publication bias was evaluated using the funnel plot and Egger’s test [14].

## 3. Results

A total of 12,826 studies were identified using keywords in the initial database search, and 103 studies were identified through the manual search of the references of the related articles. After the removal of duplicate articles and reviewing titles and abstracts, 220 full-text articles were retrieved and assessed for eligibility. During the processing process, the two reviewers had no disagreement on involving studies and a total of 19 studies met the inclusion criteria in the qualitative analysis. After failing to receive the data required to perform the analysis for one study, 18 studies were included in our final quantitative analysis (Figure 1). 

Nineteen cohort studies were included in our systematic review including 13 studies from east Asia and six studies from west Asia. No mixed population or race diversity was reported in our included studies. This included a total of 1,637,994 participants with a median age of 49.9 years (range = 31.2–71.8), a median MHO prevalence of 7.5% (0.6–24.8), a median percentage of women of 54.8% (0–77.9), and a median follow-up duration of 8.0 years (3.2–18.4). Most of the included studies used BMI as the index of obesity except for one study that used body fat [15] and another study that used waist circumference [16].

Most of the included studies defined metabolic health according to the original or revised risk factors of metabolic syndrome except for one study that used the homeostasis model to assess insulin resistance [17] and one study that defined hypertension, diabetes, and dyslipidemia according to the clinical diagnosis or use of medication [18]. The characteristics of included cohort studies were encapsulated in Appendix A.

The risk of bias assessment is summarized in Appendix A. The mean NOS score was 7.8 points with a standard deviation of 1.0 points. 

### 3.1. Primary Outcome

A total of 18 studies were included in our quantitative analysis. Since Doustmohamadian et al. [16] only reported all-cause mortality, we finally pooled 17 studies for evaluating the primary outcome. In the pooled analysis, participants with MHO had a significantly higher risk of CVD (OR, 1.36; 95% CI, 1.13–1.63; I^2^ = 75%; between-study variance (τ^2^) was 0.0859; forest plot shown in Figure 2). Considering the heterogeneity of different comparison groups (MHNW or MHNO), we performed a subgroup analysis. Participants with MHO were at a significantly higher risk of CVD than MHNW participants (OR, 1.61; 95% CI, 1.24–2.08; I^2^ = 73.%, τ^2^ = 0.12), but there was no statistically significant difference compared with MHNO participants (OR, 1.04; 95% CI, 0.80–1.36; I^2^ = 68%, τ^2^ = 0.06). The forest plot of the subgroup analysis is presented in Figure 3.

The meta-regression analysis revealed a nonsignificant and weak trend for declining CVD risk of MHO with an increasing proportion of women, age, and proportion of smokers (*p* = 0.58, *p* = 0.50, *p* = 0.60, respectively; bubble plots are shown in Figure 4, Appendix A, respectively). 

Each bubble represents one study, and the bubble size represents the study’s sample size. The regression line shows a nonsignificant and weak trend for the declining risk of CVD with the increasing proportion of women (*p* = 0.48).

Due to the good quality of our included studies, we did not omit articles for the sensitivity analysis. A sensitivity analysis was conducted by systematically omitting each study, and it showed no influential studies. This confirmed the stability of our results (forest plot shown in Appendix A). Another sensitivity analysis excluding five studies with a mean follow-up duration less than five years yielded robust results and a higher point estimate than all studies taken together (OR, 1.42; 95% CI, 1.04–1.94; Appendix A).

The funnel plot was symmetrical on inspection (Appendix A), and the Egger’s test indicated no publication bias (*p* = 0.29).

### 3.2. Secondary Outcome

There were six studies that assessed the all-cause mortality rate in our pooled analysis. This showed that participants with MHO had a significantly lower risk of all-cause mortality than the comparison groups (OR, 0.88; 95% CI, 0.78–1.00; I^2^ = 46%, τ^2^ = 0.01; Appendix A). Additionally, a subgroup analysis was performed. Participants with MHO had a significantly lower risk of all-cause mortality (OR, 0.83; 95% CI, 0.78–0.88; I^2^ = 9%, τ^2^ = 0) than the MHNO participants but a borderline significantly higher risk of all-cause mortality than the MHNW participants (OR, 1.30; 95% CI, 0.99–1.72; I^2^ = 0%, τ^2^ = 0). Heterogeneity in both subgroups was zero, and the subgroup analysis is shown in Figure 5.

We performed a sensitivity analysis of MHO and all-cause mortality by omitting each study in turn (Appendix A). An additional sensitivity analysis was performed by excluding studies with a mean follow-up duration of fewer than five years. These analyses yielded inconsistent results and revealed a trend towards high all-cause mortality (OR, 0.98; 95% CI, 0.77–1.26; Appendix A). The funnel plot of all-cause mortality was symmetrical on inspection (Appendix A), and the Egger’s test indicated no publishing bias (*p* = 0.10). 

## 4. Discussion

Similar to a previous study [7], we found that MHO increased the risk of CVD and all-cause mortality compared with MHNW in the Asian population. This result was more striking for CVD than for all-cause mortality. However, CVD risk was not significantly different between participants with MHO and MHNO participants, and MHO had an only slightly harmful all-cause mortality. Age, sex, and smoking were nonsignificant but had potential weak effective modifiers of the risk of CVD in MHO.

In view of unexpectedly lower risk of all-cause mortality of MHO than the comparison groups, we performed the subgroup analysis. The result showed that when the comparison group was MHNW, MHO had a marginally higher risk of all-cause mortality. However, when compared with MHNO, MHO had a lower risk of CVD and all-cause mortality. To analyze this comparison group, six studies selected MHNO as the comparison group. One study used waist circumference [19], and another study used body fat as indices of obesity [15]. In contrast to the different categories obtained using BMI, only the dichotomous categories of obesity and non-obesity were obtained using waist circumference and body fat indices. Considering that MHO individuals are often only mildly obese [20], waist circumference and body fat might not be sensitive enough to distinguish between MHNW and MHNO. However, BMI alone may not accurately indicate MHO [21]; the combination of these obesity indices may be more suitable to investigate the incidence of MHO than each indicator alone. For example, one Japanese study classified CVD outcomes according to BMI and glucose tolerance status [22]. In one Korean study, subjects were enrolled from a health screening program in which 60% of the participants were employees with a mean age of 40.2 years. Therefore, the selection method may have led to selection bias and the healthy worker effect [23]. Another study enrolled subjects from a health examination and did not adjust for chronic disease in their models, and therefore, possible residual confounders are likely in this study [18]. Additionally, another study included a short follow-up duration of four years, and an inverse causal relationship should be considered [24]. Other than these possible selection biases, residual confounders, and possible inverse causal relationships within studies, the ‘obesity paradox’ (obesity bring counterintuitively protective in certain conditions, the phenomenon was particularly strong in the overweight and mildly obese individuals or the MHO group) [25] was observed in our all-cause mortality results. This can be explained by a low lean body mass, but not low-fat mass, among the low BMI ranges [26].

The Asian diet is relatively high in carbohydrates. As the diet habit westernized, the Asians consumed more highly processed foods than the traditionally complex carbohydrates, which are linked to insulinemic responses and further risk of obesity and CVD [27]. At similar BMIs, Asians are reported to have more stable, atherogenic visceral fat and less subcutaneous fat than Caucasians [28]. Visceral fat has been related to macrophage infiltration, pro-inflammatory cytokines, greater vascularization, and more thrombogenic proteins which are associated with a greater metabolic risk than subcutaneous fat [28]. Thus, increased CVD risk is associated with lower BMI in Asians than in Caucasians [29]. Although a previous meta-analysis revealed sex and age to be significant effect modifiers [7], we found no significant effect, and further studies may be warranted. Sex-related differences in CVD risk could be explained by variations in visceral adipose tissue accumulation, deposition, and distribution which have been associated with insulin resistance and metabolic risks [30,31].

Since obesity is increasing in Asia, the results of this study are relevant for future studies. After choosing the reasonable comparison group and defining obesity among a heterogeneous population [32], we showed that MHO increased CVD risk and all-cause mortality. Our findings have clinical implications and define the methodological requirements of future studies. 

The present study has several limitations. First, none of our included articles adopted the latest definition of MHO. A new robust definition has been published in the last two months [33], but, to our knowledge, no article has used this definition. Second, studies that used body fat distribution and waist circumference were eligible for inclusion. However, most included articles used BMI as the only indicator of obesity. Assessment of the definition of obesity within MHO may be valuable. Third, we included studies from Northeast and West Asia. Obesity and CVD are emerging issues in South Asia [34], and further studies detailing MHO in South Asia are needed. 

## 5. Conclusions

Asian individuals with MHO had a higher risk of CVD and all-cause mortality than MHNW individuals. Future studies should be careful in selecting the index of obesity and the control group. Our results suggest that Asians, notably MHO individuals, should aim to have a normal weight to avoid CVD.

## Figures and Tables

**Figure 1 ijerph-17-01320-f001:**
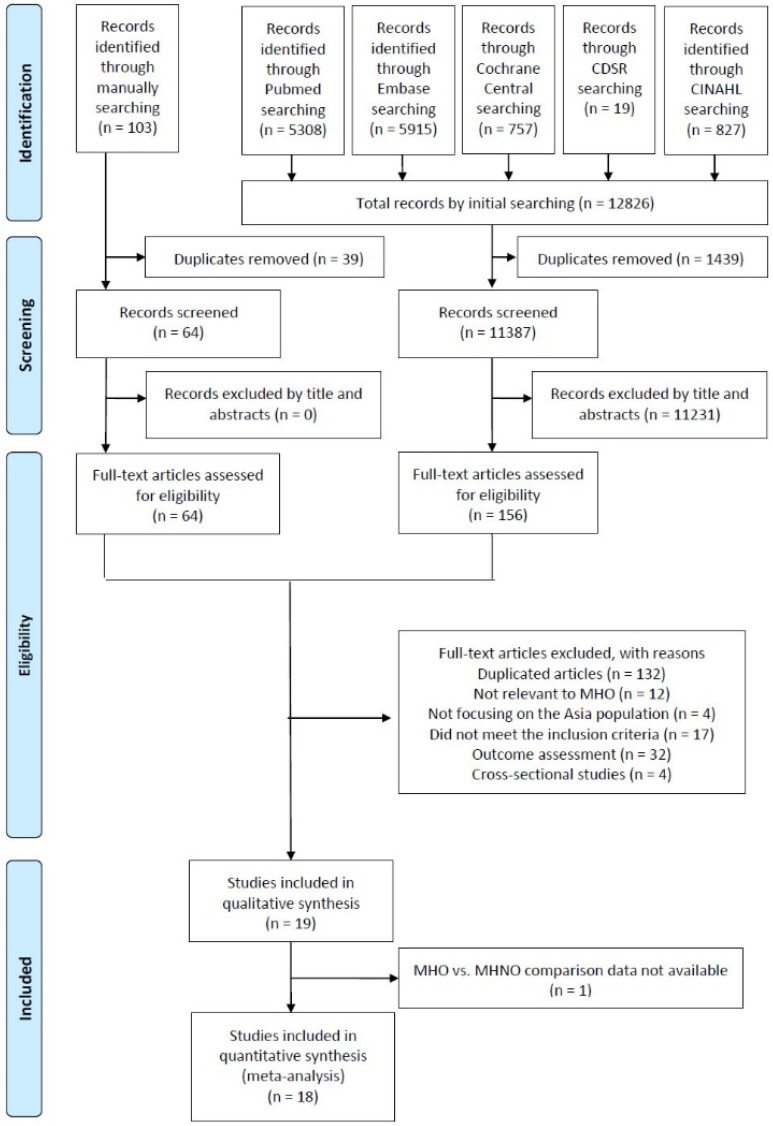
Flowchart of the study selection process. Abbreviations: CDSR: Cochrane Database of Systematic Reviews; CINAHL: Cumulative Index to Nursing and Allied Health Literature; MHO: Metabolically healthy obesity; MHNO: Metabolically healthy non-obesity.

**Figure 2 ijerph-17-01320-f002:**
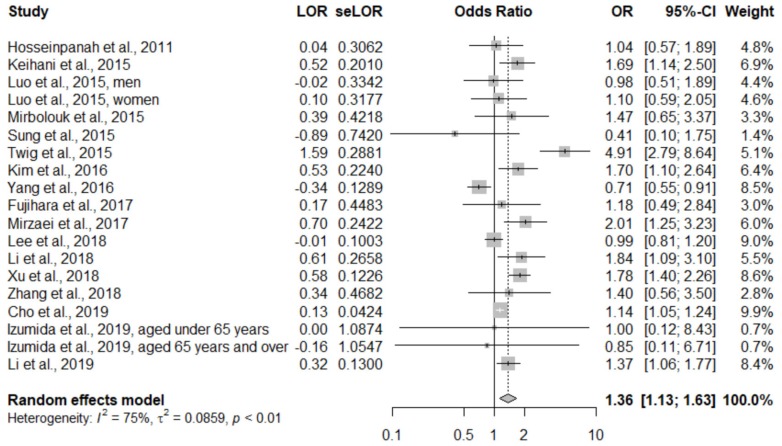
Forest plot of cardiovascular disease, comparing participants with metabolically healthy obesity and the comparison group. CI: Confidence interval; LOR: Logarithms of the odds ratio; OR: Odds ratio; se: Standard error.

**Figure 3 ijerph-17-01320-f003:**
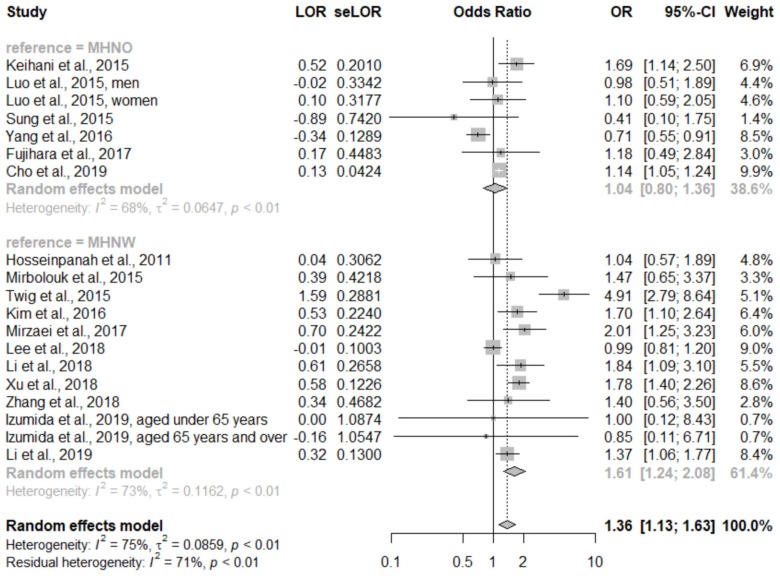
Forest plot of cardiovascular disease, comparing participants with metabolically healthy obesity and metabolically healthy non-obesity (subgroup analysis according to comparison group). CI: Confidence interval; LOR: Logarithms of the odds ratio; MHNO: Metabolically healthy obesity; MHNW: Metabolically healthy normal weight; OR: Odds ratio; se: Standard error.

**Figure 4 ijerph-17-01320-f004:**
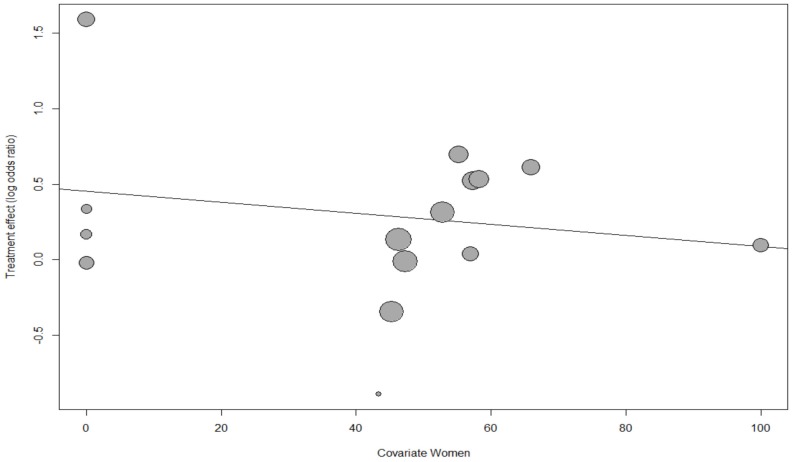
Meta-regression bubble plot of the correlation between the log odds ratio of cardiovascular disease and the proportion of women.

**Figure 5 ijerph-17-01320-f005:**
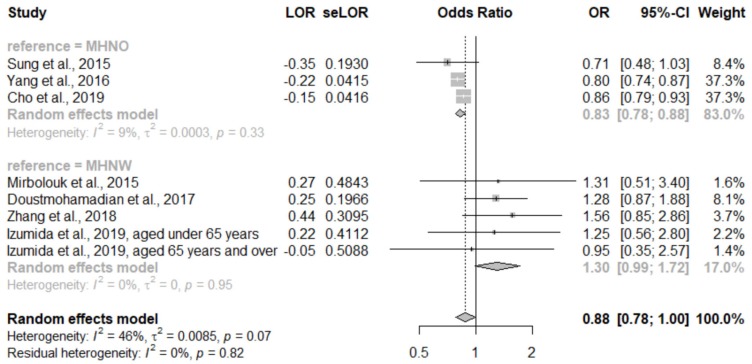
Forest plot of all-cause mortality comparing participants with metabolically healthy obesity and participants with metabolically healthy non-obesity (subgroup analysis according to comparison group). CI: Confidence interval; LOR: Logarithms of the odds ratio; MHNO: Metabolically healthy obesity; MHNW: Metabolically healthy normal weight; OR: Odds ratio; se: Standard error.

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
