# Peer review of "The Association between Metabolically Healthy Obesity, Cardiovascular Disease, and All-Cause Mortality Risk in Asia: A Systematic Review and Meta-Analysis"

_ijerph, 2020, doi:10.3390/ijerph17041320_

Round 1

Reviewer 1 Report

In general, I think the topic is interesting, the study is done in an appropriate way, and the paper deserves publication.

Before accepting the paper for publication however, I would suggest the authors to elaborate the introduction section. The authors (convincingly) explain why this additional study is needed, but do not give information on theoretical insights and/or previous research findings on the relation between obesity and cardiovascular diseases and on factors that might moderate this relationship. I think the moderator analyses of the authors make sense, but the study could be made stronger by arguing that these moderator variables are indeed the (only) relevant ones to be studied. Also in the discussion section, the authors hardly compare the results to what was found in previous research. For instance, I found the finding that the risk of all-cause mortality is significantly lower in obese people surprising, but not a lot of attention is given to this finding (how can this be explained, and is this in line with previous study findings?).

Besides that, the paper sometimes miss clarity and information. My specific comments below give more details on what sentences were unclear to me.   

Detailed comments:

line 33-35: by the way the sentence is structured, it is not very clear what exactly the comparison groups are. The description in lines 113-114 is cleared. Moreover, I find the concept ‘subgroup analyses’ confusing here because the word subgroup refers to a subgroup of studies, but in the same sentence you talk about multiple comparison groups. It is also not clear why subgroup analyses is stated in plural. Probably the authors mean that one subgroup analysis is done for the CDVr outcome variable, and one for the all-cause mortality rate. On line 29, ‘subgroup analyses’ should be written as ‘subgroup analysis’. In sum, without having read the paper, the abstract remains unclear at various places.

line 36: it is not clear what the +/- 1.0 means here

 line 49: “The Asian population 49 accounts for more than 60% of the global population.” I would drop this sentence, or replace and reformulate it (it interrupts the ‘story’ and it is not clear what population is referred to).

line 57: “it has been described as …, or to describe …”. This sentence is grammatically not correct

line 66: “…only dichromatic to obese or non-obese, including overweight and underweight individuals were excluded …”. Also this sentence is not well formulated

line 82 (& line 122): “and scanned the reference lists of primary studies manually”. it is not clear what primary studies are referred to? The description and Figure 1 suggest that reference lists of all studies retrieved from the databases (12,826 studies) were screened, but this is not made explicit.

line 86: the first inclusion criterion states that only studies with Asian participants were included. I would suggest including a clarification of what was (or would have been) done with studies with a mixed population, or with multiple populations.

line 93: it is unclear whether two reviewers decided independently from each other on the inclusion of studies. And was this done for all studies, and what was the amount of agreement? It would be good to give an idea of the reliability of this selection process, as well as of the coding process.

line 107 and following: I miss information on the R-package that was used and on the estimation procedure and options. It is also not made clear what data formats are found (maybe this should rather be added to the results section), and how these formats are converted to a common format. I guess the analysis was done on log odds ratio’s, but this is not mentioned explicitly. It is also not clear whether only one effect size is found in each study, and in case multiple effect sizes were observed in some studies, how the dependence in effect sizes is handled.

line 111: “A p-value of the Q test <0.05 or I2 >50% indicated significant heterogeneity”. A p-value says something about the statistical significance, I² something about the relative amount of heterogeneity. The word ‘significant’ therefore seems to be used in a double sense, which is confusing.

line 128: it is not clear that this is a note on the figure. I would also use ‘:’ instead of ‘,’.

line 143: what about the 18th study?

line 144: I would also say something about the size of the between-study variance

line 145: “Considering the heterogeneity of different comparison groups, …”. Unclear.

Figure 2: I would replace the abbreviation ‘TE’ for treatment effect by ‘LOR’ for Log Odds Ratio. The reason is that TE does not give an idea of what measure was used, and also the Odds Ratio mentioned in another column is a measure of the treatment effect. Moreover, there is no treatment in this study.

line 159: I would not only say that the trends are not-significant (which is not surprising, given the small number of studies), but also that the trends are weak (the regression coefficients are small, and you can also not see a pattern in the scatter plots)

line 168: systematically omitting studies is not primarily a way of detecting outlying studies, but rather a way of detecting influential studies

line 174: ‘publication bias’ instead of ‘publishing bias’

line 199: “MHO even became protective in all-cause mortality”. I think a somewhat more prudent conclusion should be drawn (this association was only found for one of the two types of comparison groups; moreover, merely an association was found, whereas the word ‘protective’ suggests a causal interpretation)

Reviewer 2 Report

The manuscript presented by Huang et al. Is an interesting study, which narrows very interesting information regarding cardiovascular health and obesity. In general, the manuscript is very well written, and the authors used an appropriate methodology. I also consider that this manuscript will be of interest to researchers, especially from Asia. However, I have the following suggestions.

Minor comments:

1. The authors in the methodology section should be more specific regarding the criteria used to select the manuscripts.
2. The figures have very small letters, it was difficult for me to read the information. I suggest improving the resolution of the figures.
3. It would be interesting to include a brief paragraph in the discussion regarding specific effects of dietary components on the development of obesity and cardiovascular disease. For example saturated fatty acids, sugar, etc.

Author Response

Response to Reviewer 2 Comments

Point 1: The authors in the methodology section should be more specific regarding the criteria used to select the manuscripts.

Thanks for your expert opinion. We revised our methods as following:

We selected articles that compared CVD risk and all-cause mortality of MHO with MHNO and MHNW in the Asian population. We searched all article types except letters and editorials for ascertaining the quality of the methodology and did not limit the language during the search. Studies were included if they met the following criteria: (1) The study cohort only recruited Asian participants who were 18 years or older; (2) The research interest was MHO; (3) Comparisons were made with MHNO and MHNW participants; (4) The study outcomes met the composite definitions of CVD (including coronary heart disease, stroke, and heart failure) and all-cause mortality; and (5) The study adopted a cohort study design.

Point 2: The figures have very small letters, it was difficult for me to read the information. I suggest improving the resolution of the figures.

Thanks for your expert opinion. We replaced the figures with larger size and proper resolution.

Point 3: It would be interesting to include a brief paragraph in the discussion regarding specific effects of dietary components on the development of obesity and cardiovascular disease. For example saturated fatty acids, sugar, etc.

Thanks for your expert opinion. We added these sentences in the discussion:  The Asian diet is relatively high in carbohydrates. As the diet habit westernized, the Asians consumed more highly processed foods than the traditionally complex carbohydrates, which are linked to insulinemic responses and further risk of obesity and CVD .

Reviewer 3 Report

This purpose of this systematic review and meta-analysis was to investigate the relationship between MHO and CVD and all-cause mortality in the Asian population and detect any differences with two control groups: MHNW and MHNO. Overall, this paper was well-conceptualized and reported, and a joy to read. Please see comments to follow for areas of improvement or clarification.

Abstract: line 40 - the authors report ORs for all-cause mortality subgroup analyses, and should similarly report ORs for CVD compared with the MHNW group. 

Introduction: although all-cause mortality is reported as the secondary outcome, I would also include some citations in the introduction. I may also lean towards including this in the title?

Methods: 

Line 80 - Systemic should say Systematic. Line 84 - while it is true that the authors investigate comparisons with the MHNW group only in the subgroup analyses, I would also include these here to be consistent with the study objectives reported at the end of the introduction. Line 86 - were the ethnicity demographics in all included studies 100% Asian? If not, how was this accounted for in the analyses? Line 90 - the authors mention that only studies using a cohort design were included, however in Line 85, they report that they searched all articles types except letters and editorials. Was there a reason study design was not embedded in the search strategy? Line 93 - the authors implicitly state that two reviewers were involved; I would like more detail on this involvement (e.g., did the two independent reviewers independently screen titles and abstracts as well as full-text papers?) Line 96 - the authors state "We extracted the following information from each study". Please provide more detail on who was involved here as well and how.  Line 116 - the authors include risk of bias scores in the appendices. Seeing that most were of high quality, it should be mentioned that sensitivity analysis excluding studies with a high risk of bias was not performed due to these reasons, or others. 

Results: line 192 - CVD should say all-cause mortality, if I am not mistaken.

Discussion:

Line 216 - please expand on the obesity paradox.  Line 226 - disposition should say deposition, if I am not mistaken.  Line 229 - can the authors outline what they mean by "correct" comparison group? Can the authors make such a statement given that, as they addressed in their limitations section, none of the included papers adopted the latest definition of MHO? 

Thank you for the opportunity to review this study. 

Author Response

Response to Reviewer 3 Comments

Point 1:  The authors report ORs for all-cause mortality subgroup analyses, and should similarly report ORs for CVD compared with the MHNW group. 

Thanks for expert opinion. We added these sentences to our abstract:

 Subgroup analyses revealed participants with MHO had a significantly higher CVDr than MHNW participants (OR=1.52; 95% CI=1.24–1.86; I2=61%), but there was no significant difference compared with MHNO participants (OR, 1.05; 95% CI, 0.81–1.34; I2 = 63%).

Point 2: Although all-cause mortality is reported as the secondary outcome, I would also include some citations in the introduction. I may also lean towards including this in the title?

Thanks for expert opinion. We amended our title to The association between metabolically healthy obesity and cardiovascular disease and all-cause mortality risk in Asia: A systematic review and meta-analysis

Point 3: Systemic should say Systematic.

Thanks for expert opinion. We amended systemic to systematic.

Point 4: While it is true that the authors investigate comparisons with the MHNW group only in the subgroup analyses, I would also include these here to be consistent with the study objectives reported at the end of the introduction.

Thanks for expert opinion. We revised the Methods as following:

We selected articles that compared CVD risk and all-cause mortality of MHO with MHNO and MHNW in the Asian population.

Point 5: Were the ethnicity demographics in all included studies 100% Asian? If not, how was this accounted for in the analyses? the authors mention that only studies using a cohort design were included, however in Line 85, they report that they searched all articles types except letters and editorials. Was there a reason study design was not embedded in the search strategy? 

Thanks for expert opinion. We revised the Methods as following:

We searched all article types except letters and editorials for ascertaining the quality of the methodology and did not limit the language during the search. Studies were included if they met the following criteria: (1) The study cohort only recruited Asian participants who were 18 years or older; (2) The research interest was MHO; (3) Comparisons were made with MHNO and MHNW participants; (4) The study outcomes met the composite definitions of CVD (including coronary heart disease, stroke, and heart failure) and all-cause mortality; and (5) The study adopted a cohort study design.

Point 6: The authors implicitly state that two reviewers were involved; I would like more detail on this involvement (e.g., did the two independent reviewers independently screen titles and abstracts as well as full-text papers?) 

Thanks for expert opinion. We revised the Methods as following:

Two reviewers independently screened the titles, abstracts as well as full-text papers to determine eligibility, and disagreements were resolved by discussion with a third reviewer.

Point 7: The authors state "We extracted the following information from each study". Please provide more detail on who was involved here as well and how.

Thanks for expert opinion. We added these sentences to the Methods:

from the tables and text of the full paper. Adjusted odds ratio (OR) for the risk of CVD and all-cause mortality were extracted from each study. If multiple effect sizes were reported in one study, the highest effect size was selected.

Point 8:  The authors include risk of bias scores in the appendices. Seeing that most were of high quality, it should be mentioned that sensitivity analysis excluding studies with a high risk of bias was not performed due to these reasons, or others. 

Thanks for expert opinion. We added these sentences to the Results:

Due to the good quality of our included studies, we did not omit articles for the sensitivity analysis.  

Point 9:  CVD should say all-cause mortality, if I am not mistaken.

Thanks for expert opinion. We revised the Results as following:

These analyses yielded inconsistent results and revealed a trend towards high all-cause mortality (OR, 0.98; 95% CI, 0.74–1.31; Table S6).

Point 10: Please expand on the obesity paradox.

Thanks for expert opinion. We revised the Discussion as following:

  Other than these possible selection biases, residual confounders, and possible inverse causal relationships within studies, the ‘obesity paradox’ (obesity bring counterintuitively protective in certain conditions, the phenomenon was particularly strong in the overweight and mildly obese individuals or the MHO group).

Point 11: Disposition should say deposition, if I am not mistaken

Thanks for expert opinion. We amended disposition to deposition.

Point 12: Can the authors outline what they mean by "correct" comparison group? Can the authors make such a statement given that, as they addressed in their limitations section, none of the included papers adopted the latest definition of MHO? 

Thanks for expert opinion. We revised the Results as following:

After choosing the reasonable comparison group and defining obesity among a heterogeneous population, we showed that MHO increased CVD risk and all-cause mortality.

Reviewer 4 Report

Thanks for the opportunity to review this paper. This is a well-written and very thoughtful systematic review and meta-analysis paper, which has the potential to be added to the literature.

Only minor spelling error to be corrected as listed below:

In the introduction, line 66: "dichromatic" should be "dichotomize". In the method section, line 80: "The" should be "the".

Author Response

Response to Reviewer 4 Comments

Point 1:  "dichromatic" should be "dichotomize".

Thanks for expert opinion. After merging opinions from different reviewers , we revised the sentence as following:

Furthermore, in several Korean studies that compared MHO individuals with “metabolically healthy non-obese (MHNO)”, individuals are classified as obese or non-obese, and data on overweight and underweight individuals were excluded from these meta-analyses .

Point 2: "The" should be "the"

Thanks for expert opinion. We amended "The" to "the".

Round 2

Reviewer 1 Report

The authors replied to most of my minor comments, and made (acceptable to satisfying) changes in the manuscript.

However, there was no reply to my major comment, and I do not see substantial changes in the manuscript to account for this comment. I therefore repeat this comment:

"Before accepting the paper for publication however, I would suggest the authors to elaborate the introduction section. The authors (convincingly) explain why this additional study is needed, but do not give information on theoretical insights and/or previous research findings on the relation between obesity and cardiovascular diseases and on factors that might moderate this relationship. I think the moderator analyses of the authors make sense, but the study could be made stronger by arguing that these moderator variables are indeed the (only) relevant ones to be studied. Also in the discussion section, the authors hardly compare the results to what was found in previous research. For instance, I found the finding that the risk of all-cause mortality is significantly lower in obese people surprising, but not a lot of attention is given to this finding (how can this be explained, and is this in line with previous study findings?)."

Regarding my smaller comments, the authors did not reply to these issues:

It would be good to give an idea of the reliability of this selection process, as well as of the coding process. I miss information on the R-package that was used and on the estimation procedure and options. (in the revision, the authors say that RStudio was used, but they still do not give the information I asked for)

Regarding my question whether some studies reported on multiple effect sizes and how this was handled, the revised manuscript now says (line 106): "If multiple effect sizes were reported in one study, the highest effect size was selected."  I strongly disagree with that approach, as it will bias the meta-analytic results towards a larger effect estimate. I am very strong about this: if this approach is not changed, the study should not be published! The simplest (and acceptable) approach is to calculate the average effect size and the average sampling variance per study before combining the effects in a meta-analysis.

Author Response

Dear Reviewer,

Round 3

Reviewer 1 Report

The revised manuscript sufficiently accounts for my previously raised concerns.